# The sea ice component of GC5: coupling SI[3] to HadGEM3 using conductive fluxes

Ed Blockley[1], Emma Fiedler[1], Jeff Ridley[1], Luke Roberts[1], Alex West[1], Dan Copsey[1], Daniel Feltham[3], Tim Graham[1], David Livings[3], Clement Rousset[2], David Schroeder[3], Martin Vancoppenolle[2]

[1]Met Office, FitzRoy Road, Exeter, EX1 3PB, UK
[2]Institut Pierre-Simon Laplace, Sorbonne Université/CNRS/IRD/MNHN, Paris, France
[3]Department of Meteorology, University of Reading, Reading, RG6 6ET, UK

*Correspondence to*: Ed Blockley (ed.blockley@metoffice.gov.uk)

**Abstract.** We present an overview of the UK's Global Sea Ice model configuration version 9 (GSI9), the sea ice component of the latest Met Office Global Coupled model, GC5. The GC5 configuration will, amongst other uses, form the physical basis for the HadGEM3 (Hadley Centre Global Environment Model version 3) climate model and UKESM2 (UK Earth System Model version 2) Earth system model that will provide the Met Office Hadley Centre/UK model contributions to CMIP7 (Coupled Model Intercomparison Project Phase 7). Although UK ocean model configurations have been developed for many years around the NEMO (Nucleus for European Modelling of the Ocean) ocean modelling framework, the GSI9 configuration is the first UK sea ice model configuration to use the new native NEMO sea ice model, SI[3] (Sea Ice modelling Integrated Initiative). This replaces the CICE (Community Ice CodE) model used in previous configuration versions. In this paper we document the physical and technical options used within the GSI9 sea ice configuration. We provide details of the implementation of SI[3] into the Met Office coupled model and the adaptations required to work with our 'conductivity coupling' approach, and provide a thorough description of the GC5 coupling methodology. A brief evaluation of sea ice simulated by the GC5 model is included, with results compared to observational references and a previous Global Coupled model version (GC3.1) used for CMIP6, to demonstrate the scientific credibility of the results.

## 1 Introduction

For over a decade, the Met Office have been developing global ocean model configurations based upon the NEMO (Nucleus for European Modelling of the Ocean) ocean modelling framework (Madec et al., 2022). Standard UK configurations (see Guiavarc'h et al., 2024; Storkey et al., 2018; Ridley et al., 2018) are developed in collaboration with partners at the National Oceanography Centre (NOC), British Antarctic Survey (BAS), and the Centre for Polar Observation and Modelling (CPOM) as part of the UK's Joint Marine Modelling Programme (JMMP). These global configurations, amongst other purposes, provide the ocean and sea ice components of the Met Office Global Coupled (GC) model, which is used for modelling across a range

of timescales, from short-range forecasting to centennial climate projections, as part of the Met Office seamless forecasting approach (Brown et al., 2012).

This paper is focussed upon the latest version of the Met Office Global Coupled configuration, GC5 (Xavier et al., *in prep*), which will form the physical basis for Met Office Hadley Centre/UK model contributions to CMIP7 (Coupled Model Intercomparison Project Phase 7) with the HadGEM3 (Hadley Centre Global Environment Model version 3) physical climate

model and UKESM2 (UK Earth System Model version 2). The GC5 coupled model is comprised of a combined Global Atmosphere and Land component (GAL9; Willett et al., *in prep*), coupled using the OASIS3-MCT coupler (Ocean Atmosphere Sea Ice Soil Model Coupling Toolkit; Craig et al., 2017) to a combined Global Ocean and Sea Ice component (GOSI9; Guiavarc'h et al., 2024). The atmosphere and land components of GC5 run on a staggered latitude–longitude grid using the Met Office Unified Model (MetUM, hereafter simply UM) and JULES (Joint UK Land Environment Simulator; Best et al.,

2011) modelling systems respectively. The ocean and sea ice components run on a tripolar grid and are built around the NEMO ocean modelling framework. The sea ice component of GC5 is the new NEMO sea ice model, $SI^3$ (Sea Ice modelling Integrated Initiative; Vancoppenolle et al., 2023). This is a change from previous model configurations, which used the CICE sea ice model (Hunke et al., 2015). The modelling systems used for the atmosphere, land, and ocean components of GC5, however, are still the UM, JULES, and NEMO respectively, consistent with previous GC model versions, e.g., the GC3.1 configuration

used for CMIP6 (Williams et al., 2017).

This paper provides an in-depth description of the "GSI9" (Global Sea Ice version 9) sea ice model configuration, which forms the sea ice component of the GC5 model. A brief evaluation of the configuration is provided using sea ice model output simulated by the fully coupled GC5 system. Documentation and wider performance of the GC5 model meanwhile are discussed in GC5 paper (Xavier et al., *in prep*), whilst a more detailed analysis of the ocean in the context of forced ocean-sea ice

experiments can be found in the GOSI9 system description paper of Guiavarc'h et al. (*2024*).

As well as documenting the GSI9 sea ice model configuration, which is built around the $SI^3$ model, this paper also describes the steps undertaken to incorporate $SI^3$ into the Met Office coupled model – including adaption of $SI^3$ to work with the "conductivity coupling" scheme used by the Met Office (West et al., 2016; Ridley et al., 2018). The coupling in the original implementation of HadGEM3 is thoroughly documented in Hewitt et al. (2011). However, with subsequent changes to the

Global Coupled model made over more than a decade, some aspects require updating. Moreover, the change from CICE to $SI^3$ means that many of the detailed schematics and processes described in Hewitt et al. (2011) are now out-of-date. We therefore provide a complete documentation of the coupling within GC5, with a particular focus on the sea ice exchanges.

This paper is organised as follows: Section 2 describes the GSI9 sea ice configuration used within GC5; Section 3 provides details on the integration of $SI^3$ within the Met Office coupled model, including modification of $SI^3$ to work with the Met

Office conductivity coupling, and presents a detailed overview of sea ice coupling within GC5; Section 4 provides a brief evaluation of GC5 sea ice output, with comparison to the GC3.1 model used within CMIP6; Section 5 ends the paper with some discussion and future plans.

## 2 GSI9 sea ice configuration

The GSI9 sea ice model configuration is based upon the native NEMO sea ice model, SI[3] (Vancoppenolle et al., 2023), which
was developed from the LIM3 model of Rousset et al. (2015) with some functionality merged from CICE. SI[3] was first made
available at NEMO version 4.0; it is fully embedded in the code and invoked from within the Surface Boundary Code (SBC)
module. The version of SI[3] used for GSI9 is based on the NEMO version 4.0.4 release as described in Guiavarc'h et al. (2024).
NEMO is the ocean modelling framework of choice for the UK and has formed the basis of global ocean model components
for over a decade. Met Office are part-owners of NEMO, and therefore SI[3], as one of two UK NEMO consortium members
(alongside NOC). Use of SI[3], therefore, offers considerable efficiencies related to management overheads, technical
development of the code, and integration into Met Office systems. For example, an advantage of using the sea ice model native
to NEMO is that the coupling is simplified; interpolation of velocity points required between NEMO (Arakawa C-grid) and
CICE (Arakawa B-grid) at previous configurations (Hewitt et al., 2011) is no longer necessary.

### 2.1 Model structure

Aside from the change in sea ice model, the sea ice physics used within GSI9 remains similar to the previous, CICE-based,
GSI8.1 configuration documented in Ridley et al. (2018). Like CICE, SI[3] is a dynamic-thermodynamic continuum sea ice
model that includes an ice thickness distribution (ITD; see Thorndike et al., 1975), conservation of horizontal momentum, an
elastic-viscous-plastic (EVP) rheology, and energy-conserving halo-thermodynamics (Vancoppenolle et al., 2023). SI[3] is run
on the same grid as the NEMO ocean model component and on every ocean time-step; the sea ice "levitates" above the
modelled ocean surface, rather than being embedded within it.

For the GSI9 configuration, five thickness categories are used to model the sub-grid-scale ITD, and an additional ice-free
category represents open water. The bounds of the thickness categories are determined using a function of domain-mean ice
thickness, which is specified as 2.0 m (namelist variable *rn_himean*). This sets the maximum thickness category bounds to
0.00 m, 0.45 m, 1.13 m, 2.14 m, 3.67 m, and 99.0 m. As with previous model versions, the ice-atmosphere exchange is
undertaken separately for each ice thickness category using a 'conductivity coupling' scheme in which surface exchanges are
calculated externally within JULES. More details on the model coupling can be found in Section 3 below.

The sea ice namelist values used in the GSI9 configuration are provided in Appendix A, with SI[3] options in Table A1 and
JULES options in Table A2. Departures from the SI[3] default options are highlighted in the tables and descriptions of these
changes are included. Full details of the GSI9 sea ice configuration are provided in the following subsections, covering
dynamics, thermodynamics, and radiation components.

## 2.2 Dynamics

Horizontal sea ice velocities are calculated by solving the momentum equation of Hibler (1979), which includes terms for internal ice stress, wind and ice-ocean stresses, sea surface tilt and Coriolis effects. The ice properties of the different thickness categories are advected following the horizontal velocity field, using the second order scheme of Prather (1986).

After the advection has been performed, mechanical deformation and lead opening converts thinner ice to either thicker ice or open water, by redistributing the global ice state variables into the different ice thickness categories. Following Thorndike et al. (1975), the redistribution function is separated into three components: (i) dynamical inputs (opening and net closing rates); (ii) the participation function, which describes the amount of ice with a given thickness participating in the mechanical deformation; and (iii) the transfer function, which determines to where in thickness space the ice is transferred as a result of

deformation. The opening and net closing rates are determined following Flato and Hibler (1995), using a formulation that includes energy dissipation by shear and convergence, and a deformation term which relates this to the rheology. The EVP rheology of Hunke and Dukowicz (2002) is used, which employs an elastic wave modification to improve the computational efficiency of a viscous-plastic (VP) rheology. Although the "adaptive" version of the EVP rheology (aEVP; Kimmritz et al., 2016) is the default in $SI^3$, this is not used in the GSI9 configuration owing to issues arising from interaction with the ice shelf

basal melt parameterisation in the ocean model (Guiavarc'h et al., 2024; Storkey et al., 2018). Super-cooled water at the edge of the Ronne-Filchner ice shelf led to continuous sea ice growth in the Weddell Sea using aEVP, where a much higher proportion of stationary ice is simulated than for EVP, associated with the improved convergence of the aEVP rheology. These issues will be addressed in future configurations. Ice strength is parameterised following Hibler (1979), which represents a departure from previous CICE configurations where the strength scheme of Rothrock (1975) was utilised.

The participation function for the mechanical redistribution is that of Lipscomb et al. (2007), which favours the closing of open water and deformation of thin ice over the deformation of thicker ice. Participation is independent of whether the deformation process is rafting or ridging. The transfer function considers rafting and ridging separately: rafting doubles the ice thickness, and newly ridged ice is linearly redistributed to the new thickness categories using a function based on Hibler (1980). Mechanical redistribution in $SI^3$ is formulated to conserve ice area, with ice volume remaining constant under rafting. Under

ridging, mass from the ocean is added to the sea ice since observations show that newly formed pressure ridges are porous (Leppäranta et al., 1995; Høyland, 2002). Additionally, during deformation, a fraction of snow falls into the ocean, set here to 50% (via namelist parameters *rn_fsnwrdg* and *rn_fsnwrft*).

Ice-ocean momentum exchange is calculated following a standard approach, using a simple bulk formula with a constant exchange coefficient and rotation angle, and using the ocean current velocity provided by NEMO. The neutral drag coefficient

has been increased for GSI9 from the $SI^3$ default of 5.0 x $10^{-3}$ to 1.0 x $10^{-2}$, consistent with the previous GSI configurations. This was partly motivated by improvements to the sea ice shown by Roy et al. (2015) when using increased drag. In order to increase the ocean time-step, semi-implicit ice-ocean drag was implemented within the ocean component of GC5 (Guiavarc'h et al., 2024). Wind stress is provided as an external forcing, calculated in JULES as part of the surface exchange scheme. A

parameterisation of ice-atmosphere form drag based on Lüpkes et al. (2012) is used within JULES, following Renfrew et al. (2019), including stability dependence and floe size as part of the form drag parameterisation (Lock et al., 2022). This scheme is different from that used in previous GSI configurations and results in a net reduction of atmosphere-ice drag, as detailed in Renfrew et al. (2019).

## 2.3 Thermodynamics

Thermodynamic growth and melt of the sea ice is modelled after the dynamics are applied, using a multilayer scheme based on Bitz and Lipscomb (1999). For each thickness category, as in previous GSI configurations, SI$^3$ models a snow-ice column consisting of four vertical layers of ice, plus an optional snow layer above. The SI$^3$ default is for two ice layers, but Vancoppenolle et al. (2023) suggests that between two and five are suitable. The thermodynamics scheme has been modified following West et al. (2016) to utilise ice surface temperatures and conductive heat fluxes into the ice provided by the surface exchange scheme in JULES. Full details are given in Section 3. Ice-ocean sensible heat flux is calculated following Maykut and McPhee (1995) as a function of local turbulent friction velocity and temperature difference between the ice and ocean, with a transfer coefficient specified as $5.7 \times 10^{-3}$. Additionally, the thermal conductivity of snow has been increased from the default 0.31 to 0.50 W m$^{-1}$ K$^{-1}$ to generate thicker ice in winter and improve summertime sea ice area and extent.

The thermodynamics also includes a lateral melting scheme that reduces the ice concentration if sea surface temperature (SST) is above freezing. The method imposes a lateral melt rate as a function of ice concentration and SST, following Bitz et al. (2001) and, unlike in previous GSI configurations, includes a parameterisation for floe size distribution. Further details are provided in Vancoppenolle et al. (2023). The default behaviour in SI$^3$ of using heat in leads for basal melting of the sea ice before heating the ocean has been turned off in GSI9 to allow ocean warming in the marginal ice zone and at the ice edge. This in turn resulted in a large increase in lateral melting, which necessitated tuning of that scheme. The beta exponent and the minimum floe diameter, used in the lateral melting scheme to describe the relationship between ice concentration and floe diameter (see equations 26-27 of Lüpkes et al., 2012), were respectively increased from the default 1.0 to the maximum recommended value of 1.2, and from the default 8 m to the maximum recommended size of 10 m. This had the effect of reducing the lateral melting and increasing the basal melting.

Once the thermodynamic melt and growth rates have been calculated, ice properties are exchanged between neighbouring thickness categories. As described by Rousset et al. (2015), this is analogous to a transport in thickness space, where the velocity is equal to the net ice growth rate and is achieved using the semi-Lagrangian linear remapping scheme of Lipscomb (2001).

Unlike in previous GSI configurations, the vertically-averaged bulk ice salinity in SI$^3$ evolves in time, for each thickness category, as a function of salt uptake during ice growth, gravity brine drainage and flushing (following Vancoppenolle et al., 2009). Salinity is assumed to have a linear vertical distribution with a profile shape dependent on the evolving bulk salinity. Salinity is used for both freshwater exchange and in the calculation of all sea ice thermodynamic properties including specific

heat, thermal conductivity, enthalpy, and freezing/melting temperatures. Full details of the salinity scheme are given by Vancoppenolle et al. (2023).

## 2.4 Radiation

Under the conductivity coupling approach used within GC configurations (see Section 3 below), the radiation is calculated
externally to the sea ice model as part of the surface exchanges in JULES. The CCSM3 (Community Climate System Model) scheme from CICE (Hunke et al., 2015) is used within JULES for computing albedo and radiative fluxes over sea ice. The scheme remains similar to that used in the previous model version described by Ridley et al. (2018), with the impact of melt ponds on albedo calculated in JULES using ponds modelled by the topographic melt pond formulation of Flocco et al. (2010, 2012) within $SI^3$. A further modification has been made here to allow the penetration of visible light into the sea ice. More
details on these aspects are provided in Section 3 below as part of the coupling documentation.

## 3 Coupling $SI^3$ within GC5

As $SI^3$ is called from within the NEMO ocean model, the atmosphere-ice coupling is inherited from the atmosphere-ocean coupling framework, OASIS3-MCT (hereafter referred to as simply OASIS). The ocean (NEMO-$SI^3$) and atmosphere (UM-JULES) models run in parallel for each coupling period, which contains several (typically between 2 and 4) ocean and
atmosphere timesteps. At each coupling instance, all ice variables required by the atmosphere are passed first to the ocean model, where they are sent to the OASIS coupler along with other ocean model variables to be remapped to the atmosphere grid. UM-JULES reads these variables, which function as the bottom boundary condition for the ensuing coupling period. UM-JULES then outputs to OASIS its own variables which are required by NEMO-$SI^3$, and these are also remapped to the ocean grid by OASIS. Upon being read by NEMO, variables required by the ice are passed to $SI^3$, functioning as the top boundary
condition for the ensuing coupling period (concurrent with that of the atmosphere). This architecture is consistent with the framework documented for the initial implementation of HadGEM3 by Hewitt et al. (2011) and used for previous GC model configurations.

The coupling framework is demonstrated schematically in Figure 1. A complete list of variables passed is shown in Table 1 (ocean to atmosphere) and Table 2 (atmosphere to ocean). Figure 1The various re-gridding methods used for the different
variables are indicated. Energy or mass flux variables are passed using conservative remapping. Where possible, second-order conservative remapping is used for increased accuracy. However, most fields contain sharp, irregular horizontal gradients and are bounded above or below by physical constraints (e.g. sea ice fraction must be between 0 and 1), properties that make second-order conservative remapping undesirable owing to the potential for over-shoot. Therefore first-order conservative remapping is used in most cases. Dynamical variables such as wind speed, for which conservation is not required, are passed
using simple bilinear remapping. Typical coupling frequencies, model time-steps and resolutions used for the GC5 model configuration can be found in Table 3.

## 3.1 Conductivity coupling in GC5

SI[3] is coupled to UM-JULES using a 'conductivity coupling' framework, in which surface variables are calculated within the atmosphere model surface exchange scheme and the thermodynamic interface between the two models is placed below the surface within the top layer of the ice-snow column. This contrasts to standard bulk formulae coupling in which this interface is placed above the ice/snow surface and the sea ice thermodynamics solves for the surface variables (i.e., temperature and energy fluxes) as well. In previous GC versions, CICE was coupled to UM-JULES using the same method, based upon the implicit coupling framework described in Best et al. (2004).

The conductivity coupling framework is used instead of the traditional framework to enable surface processes to respond more quickly to changes in the atmospheric boundary layer (West et al., 2016). By placing the surface exchange in the atmosphere model, the surface temperature and surface flux can respond instantly to changes in near-surface atmosphere conditions (and vice-versa), whereas in the standard framework there would be a delay equal to the coupling period length before either could respond (see West et al., 2016). The sea ice temperatures still experience a delayed response, but this is considered a lesser problem as these are already subject to a damped, slowed response in reality. A secondary reason for using the conductivity coupling framework is to maintain consistency with all land surface types for which UM-JULES also calculates surface variables and exchanges (Best et al., 2011). The coupling methodology is described briefly below and is discussed in more detail in West et al. (2016) and Ridley et al. (2018).

In the conductivity coupling framework, the surface exchange calculations over sea ice are carried out in the atmosphere model using the implicit scheme of Best et al. (2004), and not within the sea ice models SI[3] or CICE. The following four energy fluxes are output from the surface exchange and sent through the OASIS coupler to NEMO-SI[3] to be used as the forcing for the sea ice thermodynamics:

1. Top conductive flux: flux of conduction from the surface of the snow-ice column to the middle of the top thermodynamically active layer
2. Penetrating solar flux: flux of penetrating solar radiation from the surface of the snow-ice column to the middle of the top thermodynamically active layer
3. Top melting flux
4. Net sublimation flux

The sea ice thermodynamics scheme then solves for new temperatures in the ice and snow layers using this forcing but does not solve for surface variables. At the end of each coupling period, the temperature and effective conductivity (conductivity divided by half the layer thickness) of the top thermodynamically active layer are passed through OASIS for the atmosphere to use as bottom boundary condition for the surface exchange in the ensuing coupling period.

The conductivity coupling framework used is similar to that employed by the CMIP6 GC3.1 configuration (Ridley et al., 2018) and in the initial implementation of HadGEM3 described by Hewitt et al. (2011) (using a much more basic zero-layer

thermodynamic configuration). However, its implementation in GC5 differs in a number of ways to the documentation

provided in Hewitt et al. (2011) as described below:

1. All variables relating to sea ice are now passed separately for each thickness category. This enables the surface exchange scheme to make full use of the ice thickness distribution, allowing better simulation of rapid sea ice growth in areas of thin ice (for example, as documented by Holland et al., 2006).

2. The use of multilayer thermodynamics in GC5 entails the passing of two additional variables from ocean to
225 atmosphere: the temperature and effective conductivity of the top layer of the snow-ice column. If snow thickness is zero, the top layer is taken to be the top ice layer; as snow thickness increases from 0 to a threshold $hs\_min$, the top layer temperature and conductivity passed to the atmosphere changes linearly from the values in the top ice layer to those in the snow. Note that this is an update from GC3.1 (Ridley et al. (2018), where quantities used changed abruptly from the top ice layer to a snow layer as snow thickness crossed the threshold $hs\_min$.

3. GC5 uses semi-implicit coupling to pass the four atmosphere-ice energy fluxes described in section 3.1. In this formulation, UM-JULES does not pass grid-box-mean fluxes to the ocean. Instead, it divides these by ice concentration upon receiving the new values from NEMO-SI[3] to create 'pseudo-local' fluxes. These fields are passed through OASIS to NEMO-SI[3] where they are multiplied by the same ice concentration field. The resulting grid-box-mean fluxes are provided to the sea ice model for use over the ensuing coupling period. This formulation is necessary
to both globally conserve energy and force the ice thermodynamics with energy fields proportional to the amount of ice in each grid cell. A full description of, and justification for, the semi-implicit coupling approach is given in Ridley et al. (2018).

4. The radiative melt-pond scheme used by GC5 entails the passing of additional variables in each direction. Surface temperature is passed from the atmosphere to the ocean to be used in the melt-pond scheme to determine growth/melt
of pond refrozen lids, whilst melt-pond effective area fraction (the fraction of sea ice covered by radiatively active melt-ponds) and melt-pond depth are passed from ocean to atmosphere to be used in the radiation scheme for calculating albedo.

5. Penetrating solar radiation is now modelled. Hence, in addition to the three atmosphere-ice fluxes previously included (top conductive flux, top melt flux, and net sublimation flux), a fourth energy flux, solar radiation penetrating the sea
ice surface, is passed from the atmosphere to the ocean. The proportion of penetrating solar radiation is calculated as an extension of the Semtner (1976) scheme used in previous GC configurations (Ridley et al., 2018). Visible light that penetrates the sea ice and is not scattered back out is passed through the coupler to be used in the sea ice model.

Several other sea ice variables are passed to the atmosphere beyond those already discussed: ice and snow thickness, which
are used in the albedo calculations; ice area, which is fundamental in quantifying the contribution of the sea ice surface exchange to the whole grid cell; and combined ice and ocean velocity, which is used both in dynamic boundary layer calculations and in calculating turbulent fluxes in the surface exchange.

## 3.2 Adaption of SI³ for use with conductivity coupling

As detailed above, conductivity coupling has been used with all previous GC versions, for which CICE was adapted to work
with this method. To implement conductivity coupling in SI³ for GC5, two major modifications were required. Firstly, the
NEMO coupling interface has been changed to allow the top conductive flux to be received from the OASIS coupler and used
as the top boundary condition for the thermodynamic solver, rather than the surface exchange boundary conditions
(downwelling radiative fluxes, air temperature, specific humidity). Likewise, the coupling has been modified to send the new
ice-atmosphere variables (temperature and effective conductivity) from the topmost thermodynamically active layer through
the OASIS coupler to be used in the surface exchange calculation. Secondly, the thermodynamic routine (*icethd_zdf_bl99*)
was modified to solve for only internal snow and ice layer temperatures, leaving the surface temperature equation to be
calculated elsewhere. This option is controlled in SI³ by a logical (*ln_cndflx*) within the surface boundary condition namelist.
In addition, there is an option to 'emulate' the conductivity coupling approach in cases where no external surface exchange
scheme is available (e.g., when testing, or running NEMO-SI³ in forced-atmosphere mode). This is controlled by an additional
namelist logical (*ln_cndemulate*), where SI³ calculates the conductivity fluxes needed for the surface boundary condition from
the usual bulk formulae input using its own surface exchange calculation.

## 3.3 Assessment of energy conservation in the GC5 coupling

To compare energy conservation across the coupler in GC5, global area averages of top conductive flux, top melt flux and net
sublimation flux sent to the OASIS coupler were compared to global area averages of the re-gridded flux fields received by
the ocean. Over the course of a 1-day simulation, average errors were of the order $2 \times 10^{-3}$ Wm$^{-2}$ for top conductive flux, $2 \times 10^{-5}$ Wm$^{-2}$ for net sublimation flux, and $5 \times 10^{-6}$ Wm$^{-2}$ for top melt flux, approximately 0.05%, 0.005%, and 0.05% of the
absolute flux fields respectively. These errors are similar in magnitude to those reported in Section 4 of Hewitt et al. (2011)
for HadGEM3-AO, as well as for the GC3.1 CMIP6 configuration (not shown).

## 4 Model evaluation

In this section we present a brief evaluation of the sea ice simulated by the GC5 coupled model using the above-documented
GSI9 sea ice model configuration. The intention is not to provide a thorough assessment of the sea ice performance in GC5
but rather a sanity check that the configuration documented here is performing sensibly. As per previous coupled model
versions, GC5 has been developed with traceable science across a hierarchy of model resolutions, with ocean (atmosphere)
resolution ranging from 1° (130 km) to 1⁄12° (25 km), (e.g., Guiavarc'h et al., 2024; Storkey et al., 2018; Roberts et al., 2019).
The sea ice model options used are identical across all the different model resolutions and so we limit our attention here to the
medium resolution model, which uses eORCA025 (nominal 1/4°) ocean-sea ice resolution and N216 (~60 km in midlatitudes)
atmosphere-land resolution.

The simulated sea ice evaluated in this section is the last 50 years from 100-year "present-day" control runs forced by greenhouse gases and aerosols from the year 2000 (see Williams et al., 2017). Simulated sea ice from GC5 is compared against the GC3.1 CMIP6 configuration (Williams et al., 2017; Ridley et al., 2018), along with reference datasets including observations of sea ice concentration from the Hadley Centre Sea Ice and Sea Surface Temperature dataset version 2 (HadISST.2.2.0.0; Titchner and Rayner, 2014), observations of sea surface temperature from the ESA CCI SST L4 dataset of Good et al. (2019), and sea ice thickness from the Pan Arctic Ice Ocean Modeling and Assimilation System (PIOMAS; Schweiger et al., 2011) reanalysis, which assimilates observations of sea ice concentration but not sea ice thickness. The present-day simulations employed here represent how the climate would evolve if emissions were fixed at the year 2000 level for 100 years, which of course can differ from observed conditions for the period 1990-2009. Thus, our comparisons with observations can be considered more of a benchmark than a direct assessment.

The Arctic seasonal cycle of sea ice area (Figure 2a) demonstrates that GC5 and GC3.1 are both within 20% of HadISST.2, a criterion that has been commonly used when evaluating climate models in the context of model selection (e.g., Massonnet et al., 2012), apart from in August. The sea ice area in GC5 is closer than GC3.1 to the HadISST.2 observations apart from late spring and summer when the two are very similar. In GC5 the ice concentration is greater than GC3.1 in the marginal seas in winter (Figure 2c,e), particularly the Sea of Okhotsk, where it is now closer to HadISST.2. Meanwhile, summer concentration increases are seen in the central Arctic north of Siberia, and in the Canadian Archipelago (Figure 2b,d). When compared to HadISST.2 observations, the mean spatial pattern of ice concentration (Figure 2c,e) shows excess marginal ice cover in the Greenland Sea and reduced cover in the Labrador Sea, for both models. These biases are related to the preference of the ocean model for convective overturning in the Labrador Sea rather than the northeast Atlantic, as described by Megann et al. (2014); Guiavarc'h et al. (2024). The bias of an early Arctic summer minimum in August is present in both model configurations. Having an areal minimum for Arctic sea ice in August is not uncommon for models, Roach et al. (2020) showed that around a quarter of CMIP6 models have lower average sea ice area in August than in September. In September Arctic sea ice is still melting at the peripheries but, owing to the onset of the polar night at higher latitudes, starting to freeze-up in the centre of the pack. The evolution of September sea ice area is therefore dependent on these two competing processes. Whether September area is higher or lower than August will depend on the timing of when the Arctic transitions from net melting to net growth. It is likely that the timing of this transition is out by a few days in the model. This competition between lower latitude melting and high latitude refreeze is not captured in the extent metric, which has minimum in September for both these model configurations (see e.g. Rae et al., 2015). It is also worth noting here that uncertainty in passive microwave satellite derived observations of sea ice is considerably high in summer, where the presence of surface melt-ponds can lead to overestimation of up to 25% in concentration (see Kern et al., 2020). Therefore, a non-negligible portion of the offset between model and observations could be related to errors in the observations.

Although the spatial pattern of winter mean sea ice thickness in GC5 is similar to GC3.1 (Figure 3a,b), the ice thickness in GC5 has increased across the whole Arctic, with largest increases north of the Canadian Archipelago (Figure 3d) meaning that the spatial distribution of Arctic sea ice thickness is now more comparable to the PIOMAS reanalysis (Figure 3c). This

improvement in the central Arctic thickness pattern is associated with changes in the sea ice dynamics. The strength of the Beaufort Gyre is lower in GC5 than GC3.1 (Figure 3e,f), meaning a longer residence time, and subsequent thickening, of the multi-year sea ice piled up north of Greenland and the Canadian archipelago. This reduction in Beaufort Gyre speed is very likely linked with the change in atmosphere-sea ice drag scheme, whereby a net reduction in drag (Renfrew et al., 2019) will have led to reduced sea ice velocity, along the lines of that described by Johns et al. (2021).

The Antarctic sea ice area in GC5 has increased considerably from GC3.1 and the seasonal cycle is now comparable with HadISST.2 (Figure 4a), albeit with a slight phase-lag suggesting too-slow ice growth in early winter (May-July) and too-slow ice melt in early summer (December). It has been suggested that the suppressed rate of growth in winter is associated with the temporal pattern of insolation (Goosse et al., 2023), with other climate models displaying the same issue (e.g., DuVivier et al., 2020). Antarctic sea ice concentration has increased, and extent has expanded, in GC5 compared with GC3.1 at all times of the year, as illustrated for September and March respectively in Figure 4b,d and 4c,e. These are now much closer to the HadISST.2 dataset. This is because of a considerable improvement in GC5 of Southern Ocean surface temperatures (see Figure 4f,g and Storkey et al., 2024), where previously a warm ocean bias led to low sea ice area (Ridley et al., 2018; Rae et al., 2015). Despite the general increase in Antarctic sea ice cover, as illustrated in Fig. 4b,d there is a minor reduction in the Weddell Sea compared to GC3.1. This is associated with the emergence of multi-year sensible heat polynyas, a feature common to many climate models (Heuzé et al., 2015) including Met Office configurations (Megann et al., 2014; Ridley et al., 2022).

## 5 Conclusions

In this paper we have presented the UK's GSI9 sea ice model configuration, used within the latest version of the Met Office Global Coupled model configuration GC5, which will form the physical model basis for UK contributions to CMIP7 with HadGEM3 and UKESM. GC5 includes a change to the sea ice model component compared with earlier GC versions with the implementation of the new NEMO SI$^3$ model in place of CICE. We have described how SI$^3$ has been adapted to work with the 'conductivity coupling' used in Met Office models and provided a thorough documentation of sea ice (and wider) coupling in GC5. A brief evaluation of the GC5 sea ice using continuous year-2000 climate forcing has been presented, which shows that the sea ice simulated by this configuration compares well with observational references. A comparison was also performed with the CMIP6 model version, GC3.1, which shows that the mean state and variability of the GC5 sea ice is generally improved compared to GC3. This is particularly so in the Antarctic where the sea ice is much improved throughout the year in response to the reduction of warm biases in the ocean, as described in Storkey et al. (2024).

Future development of the Global Sea Ice configurations will include exploring the following options:

- Employing the ridging sea ice strength formulation of Rothrock (1975) and the exponential ITD transfer function of Lipscomb et al. (2007), which were used in previous GC configurations. Although these schemes have since been included in SI$^3$ (under the EU IS-ENES3 project), they were not available in time to be used in the GSI9 configuration.
- Using the land-fast ice modelling scheme of Lemieux et al. (2016), which is already available as an option in SI$^3$.

- Considering alternative sea ice rheology schemes, including the adaptive Elastic-Viscous Plastic (aEVP; Kimmritz et al., 2016) and Elastic-Anisotropic-Plastic (EAP; Tsamados et al., 2013) rheologies, which are already included in $SI^3$ (through the EU IMMERSE and IS-ENES3 projects).

- Improving the representation of ice-ocean and ice-atmosphere drag using the form-drag scheme of Tsamados et al. (2014), which has been included in JULES through the EU-APPLICATE project and is currently being ported into $SI^3$; another option for improving the ice-ocean drag is to adopt the methodology outlined in Roy et al. (2015).

- Including floe-size-distribution and wave-ice interaction as discussed by Bateson et al. (2022).

- Adapting the radiation scheme to include penetrating shortwave radiation into the sea ice under melt-ponds or simulating the freshwater impacts of melt-ponds.

## 6 Data availability

PIOMAS reanalysis data are available from the Polar Science Center webpage at http://psc.apl.uw.edu/research/projects/arctic-sea-ice-volume-anomaly/; HadISST.2.2.0.0 sea ice concentration data are available for download from the Met Office Hadley Centre at https://www.metoffice.gov.uk/hadobs/hadisst2/data/download.html; ESA CCI SST data are available from the ESA website at https://climate.esa.int/en/projects/sea-surface-temperature/data/. Owing to the size of the data sets needed for the analysis, which require large storage space of more than 1 TB, the full model output fields are not made available. They can be shared via the STFC-CEDA platform by contacting the authors.

## 7 Code availability

The NEMO ocean model and the $SI^3$ sea ice model used in GC5 are available to download from https://forge.nemo-ocean.eu/nemo/nemo/-/blob/4.2.0/README.rst, with a NEMO User Guide available online at https://sites.nemo-ocean.io/user-guide/. The CICE5 (Hunke et al., 2015) model code used here in GC3.1 is available from the Met Office code repository at https://code.metoffice.gov.uk/trac/cice/browser. Due to intellectual property copyright restrictions, we cannot provide the source code for the UM or JULES, but the UM is available for use under licence. Several research organisations and national meteorological services use the UM in collaboration with the Met Office to undertake atmospheric process research, produce forecasts, develop the UM code and build and evaluate Earth system models. To apply for a licence for the UM, go to https://www.metoffice.gov.uk/research/approach/modelling-systems/unified-model, and for permission to use JULES, go to https://jules.jchmr.org.

## 8 Author contribution

EB, EF, AW, JR, LR contributed text and/or figures/tables; EB, DC, CR, MV, AW contributed model code developments; EB, DC, TG, DL, CR, JR, LR, DS, MV, AW contributed to the model evaluation and testing; EB, DF, MV contributed to planning/conception, administration, and funding acquisition for the work; all authors contributed to interpretation of the results, reviewing draft versions of this manuscript, and to manuscript revisions following peer review.

## 9 Competing interests

The authors declare that they have no conflict of interest.

## 10 Acknowledgements

This work was developed as part of the Joint Marine Modelling Programme (JMMP), a partnership between the Met Office, National Oceanography Centre, British Antarctic Survey and Centre for Polar Observation and Modelling. The NEMO System Team and the NEMO Sea Ice Working Group are acknowledged for their role in the development and support of the NEMO-SI$^3$ model. The authors would like to thank Richard Hill for some helpful discussions regarding NEMO coupling and OASIS. EB, DC, EF, TG, JR, and AW were supported by the Met Office Hadley Centre Climate Programme funded by DSIT. EB, EF, DL, CR, MV, and AW acknowledge funding support from the European Union's Horizon 2020 research and innovation programme under grant agreement No 824084 (IS-ENES3). EB, EF, CR, and MV acknowledge funding support from the European Union's Horizon 2020 research and innovation programme under grant agreement No 821926 (IMMERSE). DF acknowledges funding support through the Copernicus Marine Environment Monitoring Service (CMEMS) SI$^3$ project under call 87-GLOBAL-CMEMS-NEMO; CMEMS is implemented by Mercator Ocean International in the framework of a delegation agreement with the European Union. The authors would like to thank the handling editor, Olivier Marti and two anonymous referees for their contribution to the peer review for this manuscript.

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

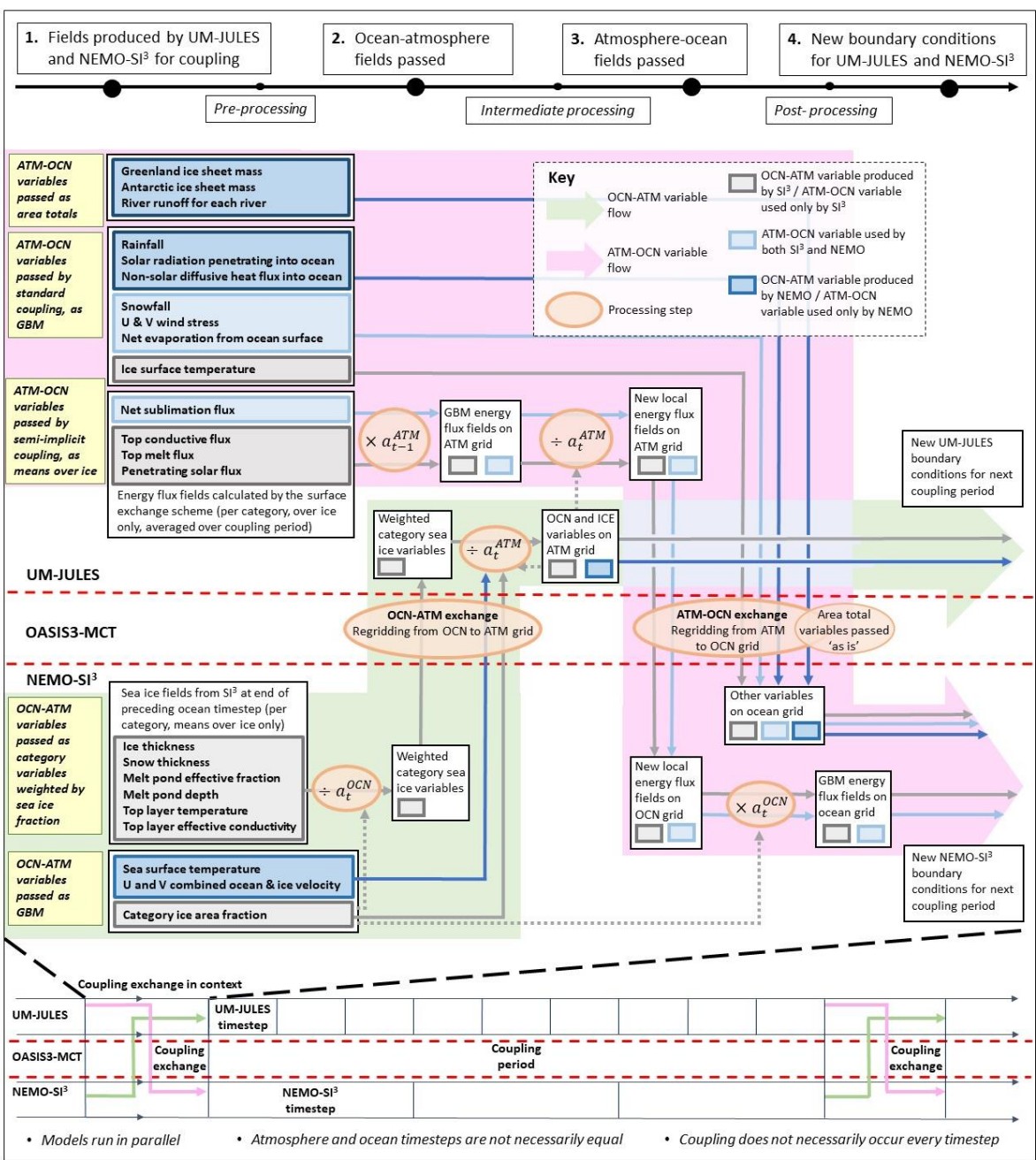

**Figure 1: Schematic overview of the coupling between UM-JULES and NEMO-SI³ used in GC5. The general coupling approach is illustrated in the form of a time-series across the bottom of the schematic; the upper 80% of the figure shows the detail for a single coupling instance, including a full list of variables passed in both directions with the key arithmetic operations performed on them.**

Here $a_t^{OCN}$ denotes the ice area fraction field sent by NEMO-SI³ at the coupling instant shown, $a_t^{ATM}$ the re-gridded ice area fraction field received by UM-JULES after being passed through OASIS, $a_{t-1}^{ATM}$ the ice area fraction field from the previous coupling instant, used by UM-JULES for the preceding coupling period. Horizontal dashed red lines are used to denote the coupling interface between NEMO-SI³ and UM-JULES via OASIS. Solid coloured arrows denote the passage of information between the various components.

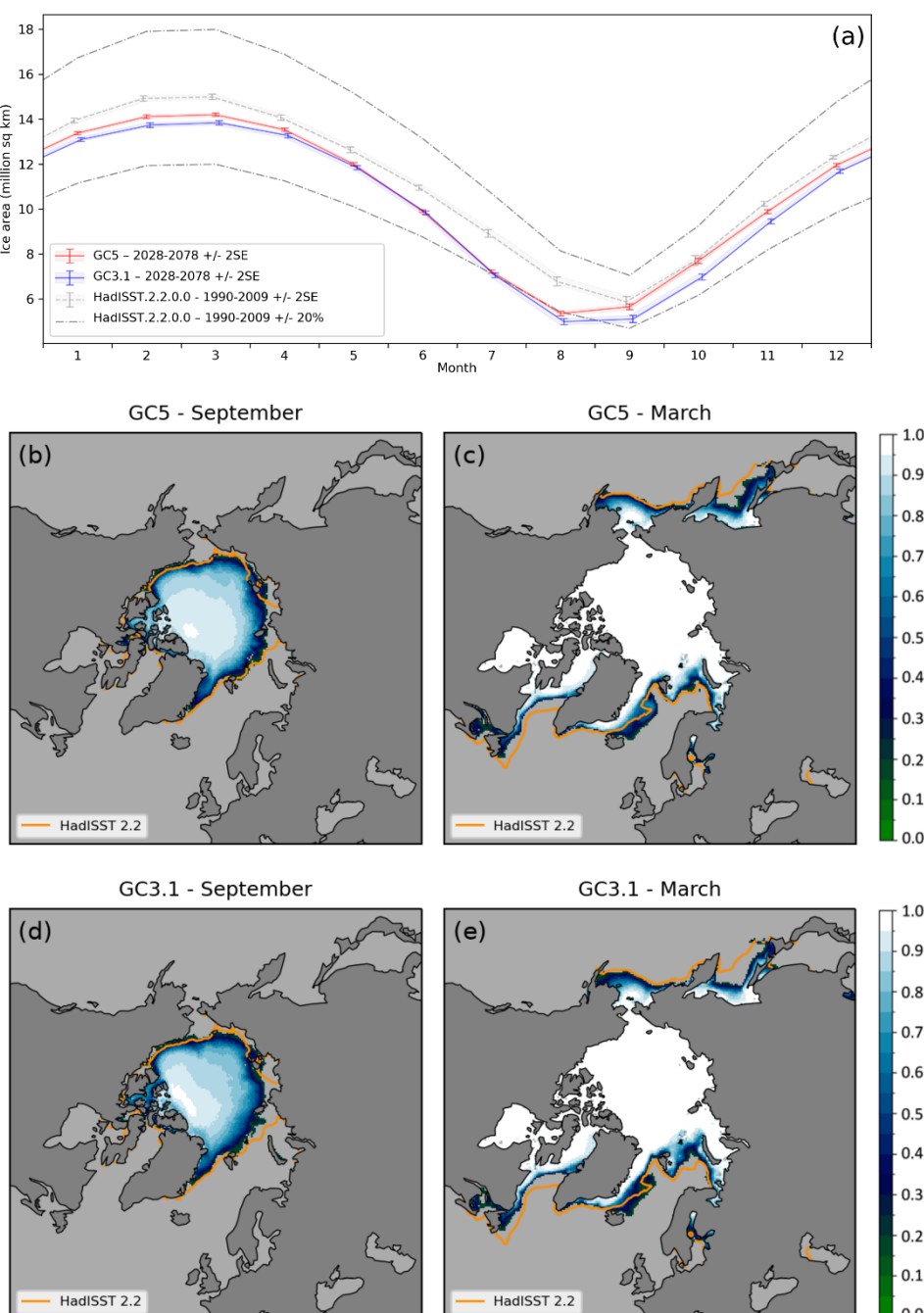

**Figure 2: (a)** The model seasonal cycle of sea ice area ($10^6$ km$^2$) in the Arctic for GC5 (red) and GC3.1 (blue). Area estimates from the HadISST.2 sea ice dataset are included in grey, with +/- 2 standard error (SE) shading and error bars, and +/- 20% indicated with chain lines. Panels (b) – (e) show simulated mean sea ice fraction with HadISST.2 0.15 contours added in orange from GC5 (b,c) and from GC3.1 (d,e) for September and March respectively. GC5 and GC3.1 data are the last 50 years from 100-year model simulations using year-2000 continuous forcing, whilst HadISST.2 data are from the period 1990-2009.

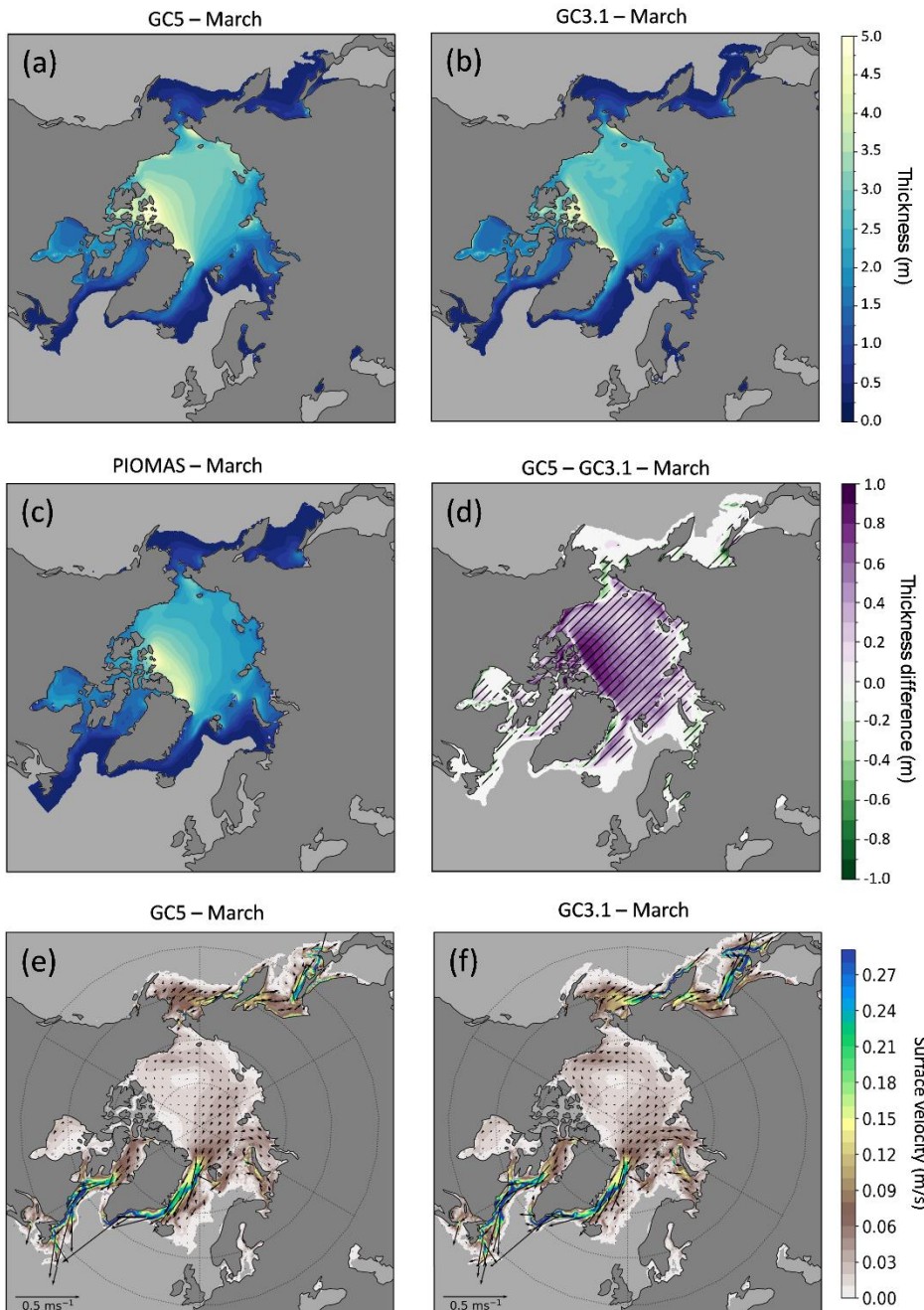

**Figure 3: March mean sea ice thickness (m) from the last 50 years of 100-year model simulations using year-2000 continuous forcing for (a) GC5 and (b) GC3.1, and (c) from the PIOMAS reanalysis for the period 1990-2009. Panel (d) shows mean sea ice thickness difference between GC5 and GC3.1 (GC5-GC3.1) with hatched areas identifying differences that are significant at the 95% level, calculated using a Welch t-test. Panels (e) and (f) show corresponding 50-year March mean sea ice velocity for GC3.1 and GC5 respectively with coloured shading depicting ice speed and velocity arrows overlain in black.**

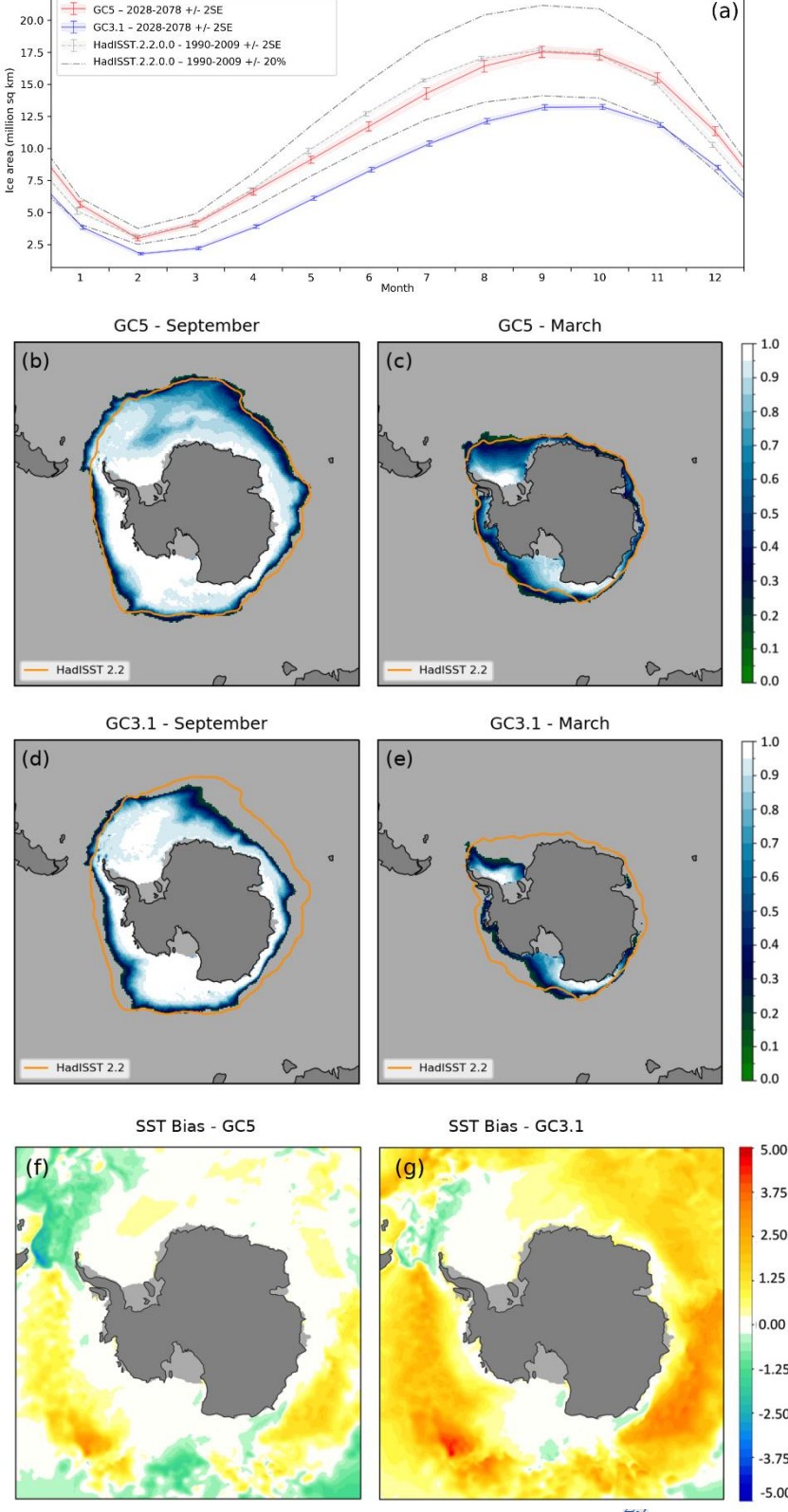

(a)

GC5 – 2028-2078 +/- 2SE
GC3.1 – 2028-2078 +/- 2SE
HadISST.2.2.0.0 – 1990-2009 +/- 2SE
HadISST.2.2.0.0 – 1990-2009 +/- 20%

Ice area (million sq km)

Month

GC5 - September     (b)     HadISST 2.2

GC5 - March     (c)     HadISST 2.2

GC3.1 - September     (d)     HadISST 2.2

GC3.1 - March     (e)     HadISST 2.2

SST Bias - GC5     (f)

SST Bias - GC3.1     (g)

| Field index | Description | Units | Remapping method |
|---|---|---|---|
| 1 | Sea surface temperature | K | First-order conservative |
| 2-6 | Temperature of the top layer of the snow-ice column | K | First-order conservative |
| 7-11 | Sea ice area fraction | 1 | First-order conservative |
| 17-21 | Sea ice thickness | m | First-order conservative |
| 22-26 | Snow thickness | m | First-order conservative |
| 27-28 | U and V components of combined ice and ocean velocity | m s$^{-1}$ | Bilinear |
| 29-33 | Melt pond effective fraction | 1 | First-order conservative |
| 34-38 | Melt pond thickness | m | First-order conservative |
| 39-43 | Effective conductivity of the top layer of the snow-ice column | W m$^{-2}$ K$^{-1}$ | First-order conservative |

Table 1: List of variables passed from NEMO-SI$^3$ to UM-JULES each coupling cycle. Note that not all field indices are documented here; field indices 12-16 are intentionally omitted as they correspond to variables that were used previously in the coupling, but which are not used in GC5.

| Field index | Description | Units | Remapping method |
|---|---|---|---|
| 45-46 | U and V atmospheric wind stresses | N m$^{-2}$ | Bilinear |
| 47 | Solar radiation penetrating into ocean | W m$^{-2}$ | First-order conservative |
| 48 | Non-solar diffusive heat flux into ocean | W m$^{-2}$ | Second-order conservative |
| 49 | Rainfall onto ocean surface (over sea ice rain drains straight into the ocean) | kg m$^{-2}$ s$^{-1}$ | First-order conservative |

| 50 | Snowfall onto sea ice and ocean surface | kg m$^{-2}$ s$^{-1}$ | First-order conservative |
|---|---|---|---|
| 51 | Net evaporation from ocean surface | kg m$^{-2}$ s$^{-1}$ | Second-order conservative |
| 52-56 | Net sublimation over sea ice | kg m$^{-2}$ s$^{-1}$ | First-order conservative |
| 59-63 | Top melt flux over sea ice | W m$^{-2}$ | First-order conservative |
| 64-68 | Surface temperature over sea ice | K | First-order conservative |
| 69-73 | Top conductive flux over sea ice | W m$^{-2}$ | First-order conservative |
| 74 | Greenland ice sheet mass | kg | None (zero-dimensional field) |
| 75 | Antarctic ice sheet mass | kg | None (zero-dimensional field) |
| 76 | River runoff for each river | kg s$^{-1}$ | None (list of single values passed) |
| 77-81 | Solar radiation penetrating into sea ice | W m$^{-2}$ | First-order conservative |

**Table 2: As Table 1 but for variables passed from UM-JULES to NEMO-SI³. NB. field indices 57-58 are intentionally omitted as they correspond to variables that were used previously in the coupling, but which are not used in GC5.**

| Configuration | Coupling frequency | Atmosphere timestep [nominal resolution] | Ocean-sea ice timestep [nominal resolution] |
|---|---|---|---|
| GC5-LL [N96-ORCA1] | 3 hours | 45 mins [250 km] | 90 mins [100 km] |
| GC5-MM [N216-ORCA025] | 1 hour | 15 mins [100 km] | 30 mins [25 km] |

**Table 3: typical coupling frequencies with corresponding model time-steps and nominal resolution used by the atmosphere-land and ocean-sea ice components for the low and medium resolution configurations of the GC5 coupled model.**

**Appendix A: GSI9 model namelists**

| SI³: namelist | | |
|---|---|---|
| Ice dynamics (namdyn) | ln_dynall = .true. | use full ice dynamics (rheology + advection + ridging/rafting + correction) |
| | ln_landfast_l16 = .false. | use Lemieux (2016) land-fast ice scheme |
| | rn_ishlat = 2.0 | lateral boundary condition for sea ice dynamics (2: no slip) |
| Advection (namdyn_adv) | ln_adv_pra = .true. | use Prather (1986) advection scheme |

| | | |
|---|---|---|
| Mechanical deformation (namdyn_rdgrft) | ln_partf_exp = .true. | use Lipscomb et al. (2007) exponential participation function |
| | ln_partf_lin = .false. | use Thorndike et al. (1975) linear participation function |
| | ln_rafting = .true. | activate ice rafting |
| | ln_ridging = .true. | activate ice ridging |
| | ln_str_h79 = .true. | use Hibler (1979) ice strength parameterisation |
| | rn_astar = 0.03 | exponential measure of ridging ice fraction |
| | rn_craft = 5.0 | coefficient for smoothness of hyperbolic tangent in rafting |
| | rn_crhg = 20.0 | parameter for Hibler (1979) ice strength |
| | rn_csrdg = 0.5 | fraction of shearing energy contributing to ridging |
| | rn_fpndrdg = 1.0 | pond fraction conserved during ridging |
| | rn_fpndrft = 1.0 | pond fraction conserved during rafting |
| | rn_fsnwrdg = 0.5 | fraction of snow volume conserved during ridging |
| | rn_fsnwrft = 0.5 | fraction of snow volume conserved during rafting |
| | rn_hraft = 0.75 | threshold thickness (m) between rafting / ridging |
| | rn_hstar = 25.0 | determines maximum ridged ice thickness (m) (Hibler, 1980) |
| | rn_porordg = 0.3 | porosity of newly ridged ice (Leppäranta et al., 1995) |
| | rn_pstar = 2.0e+4 | parameter for Hibler (1979) ice strength (N m$^{-2}$) |
| Rheology (namdyn_rhg) | ln_aevp = .false.* [default = .true.] | use adaptive EVP rheology |
| | ln_rhg_evp = .true. | use EVP rheology |
| | nn_nevp = 120* [default = 100] | number of EVP subcycles |
| | rn_creepl = 2.0e-9 | creep limit (s$^{-1}$) |
| | rn_ecc = 2.0 | eccentricity of elliptical yield curve |
| | rn_relast = 0.333 | ratio of elastic timescale to ice time step |
| Ice thickness distribution (namitd) | ln_cat_hfn = .true. | ice categories defined by function of rn_himean$^{-0.05}$ |
| | ln_cat_usr = .false. | ice categories defined by user |
| | rn_himax = 99.0 | maximum allowed ice thickness (m) |
| | rn_himean = 2.0 | domain-mean ice thickness (m) |
| | rn_himin = 0.1 | minimum ice thickness (m) |
| Generic ice parameters (nampar) | jpl = 5 | number of ice categories |
| | ln_icedyn = .true. | use ice dynamics |
| | ln_icethd = .true. | use ice thermodynamics |

| | nlay_i = 4*<br>  [default = 2] | number of ice layers |
|---|---|---|
| | nlay_s = 1 | number of snow layers |
| | rn_amax_n = 0.997 | maximum ice concentration for northern hemisphere |
| | rn_amax_s = 0.997 | maximum ice concentration for southern hemisphere |
| Surface boundary conditions (namsbc) | ln_cndflx = .true.*<br>  [default = .false.]<br>ln_cndemulate = .false. | use conduction flux as surface boundary condition emulate conduction flux |
| | nn_flxdist = -1 | redistribute heat flux over ice categories (-1: do nothing) |
| | nn_snwfra = 2 | fraction of ice covered by snow (2: CICE method, Hunke et al. (2015)) |
| | rn_cio = 1.0e-2*<br>  [default = 5.0e-3] | ice-ocean drag coefficient |
| | rn_snwblow = 0.66 | coefficient for ice-lead partition of snowfall |
| Thermodynamics (namthd) | ln_iceda = .true. | activate lateral melting |
| | ln_icedh = .true. | activate ice thickness change from growing/melting |
| | ln_icedo = .true. | activate ice growth in open water |
| | ln_iceds = .true. | activate gravity drainage and flushing (brine drainage) |
| | ln_leadhfx = .false.*<br>  [default = .true.] | heat in leads is used to melt sea ice before warming the ocean |
| Ice lateral melting (namthd_da) | rn_beta = 1.2*<br>  [default = 1.0] | coefficient for lateral melting parameter |
| | rn_dmin = 10.0*<br>  [default = 8] | minimum floe diameter (m) for lateral melting parameter |
| Ice growth in open water (namthd_do) | ln_frazil = .false. | activate frazil ice collection as a function of wind |
| | rn_hinew = 0.1 | thickness of new ice formed in open water (m) |
| Melt ponds (namthd_pnd) | ln_pnd = .true. | activate melt ponds |
| | ln_pnd_cst = .false. | activate constant ice melt pond scheme |
| | ln_pnd_lev = .false.*<br>  [default = .true.] | activate level ice melt ponds |
| | ln_pnd_lids = .true. | activate frozen lids on top of melt ponds |
| | ln_pnd_topo = .true.*<br>  [default = .false.] | activate topographic melt ponds |
| | rn_apnd_min = 0.15 | minimum meltwater fraction contributing to pond growth |

| | | |
|---|---|---|
| | rn_apnd_max = 0.85 | maximum meltwater fraction contributing to pond growth |
| Ice salinity (namthd_sal) | nn_icesal = 2 | ice salinity (2: time-varying salinity parameterisation) |
| | rn_sal_fl = 2.0 | restoring ice salinity for flushing (g kg$^{-1}$) |
| | rn_sal_gd = 5.0 | restoring ice salinity for gravity drainage (g kg$^{-1}$) |
| | rn_simax = 20.0 | maximum tolerated ice salinity (g kg$^{-1}$) |
| | rn_simin = 0.1 | minimum tolerated ice salinity (g kg$^{-1}$) |
| | rn_time_fl = 8.64e+5 | restoring time scale for flushing (s) |
| | rn_time_gd = 1.73e+6 | restoring time scale for gravity drainage (s) |
| Vertical heat diffusion (namthd_zdf) | ln_cndi_p07 = .true. | use Pringle et al. (2007) sea ice thermal conductivity |
| | ln_zdf_bl99 = .true. | use Bitz and Lipscomb (1999) heat diffusion formulation |
| | rn_cnd_s = 0.5* [default = 0.31] | thermal conductivity of snow (W m$^{-1}$ K$^{-1}$) |
| | rn_kappa_i = 1.0 | radiation attenuation coefficient in sea ice (m$^{-1}$) |
| | rn_kappa_s = 10.0 | radiation attenuation coefficient in snow (m$^{-1}$) |
| SI$^3$: hard-wired parameters | | |
| | kice = 2.034396 | thermal conductivity of fresh ice (W m$^{-1}$ K$^{-1}$) |
| | rhoi = 917.0 | density of sea ice (kg m$^{-3}$) |
| | rhos = 330.0 | density of snow (kg m$^{-3}$) |
| | zch = 0.0057 | sensible heat transfer coefficient |

**Table A1: SI$^3$ namelist and hard-wired parameters of scientific significance. Variables that are changed from the SI$^3$ defaults are highlighted with an asterisk with default values given below in brackets.**

590

| JULES: namelist | | |
|---|---|---|
| | nice_use = 5 | Number of sea ice thickness categories used in surface exchange |
| [albedos] | albicev_cice = 0.78 | Visible albedo of bare ice |
| | albicei_cice = 0.36 | Near-IR albedo of bare ice |
| | albsnowv_cice = 0.98 | Visible albedo of cold snow |
| | albsnowi_cice = 0.70 | Near-IR albedo of cold snow |
| | albpondv_cice = 0.27 | Visible albedo of melt ponds |
| | albpondi_cice = 0.07 | Near-IR albedo of melt ponds |
| | dalb_mlts_v_cice = -0.10 | Change in snow Visible albedo per °C rise in temperature |

| | dalb_mlts_i_cice = -0.15 | Change in snow Near-IR albedo per °C rise in temperature |
|---|---|---|
| | dt_snow_cice = 1.0 | Permitted range of snow temperature over which albedo changes (K) |
| | ahmax = 0.3 | Sea ice thickness (m) below which albedo is influenced by underlying ocean |
| | emis_sice = 0.9760 | Emissivity of sea ice |
| | snowpatch = 0.02 | Length scale for parameterisation of non-uniform snow coverage (m) |
| [penetrating solar] | l_sice_swpen = .true. | Switch for penetrating solar radiation being passed to the sea ice model instead of being absorbed at ice surface |
| | pen_rad_frac_cice = 0.8 | Fraction of visible light that penetrates the sea ice |
| | sw_beta_cice = 0.3 | The fraction of penetrating visible light that scatters back out |
| [Lupkes formdrag] | l_iceformdrag_lupkes = .true. | Switch for diagnostic form-drag |
| | l_stability_lupkes = .true. | Switch to include the stability dependence in the parametrization of ice form drag |
| | h_freeboard_min = 0.286 | Minimum height of freeboard |
| | h_freeboard_max = 0.534 | Maximum height of freeboard |
| | beta_floe = 1.0 | Constant in parametrization of crosswind length of floe |
| | d_floe_min = 8.0 | Minimum crosswind length of floe |
| | d_floe_max = 300.0 | Maximum crosswind length of floe |
| | ss_floe = 0.5 | Sheltering constant |
| | ce_floe = 0.222 | Effective resistance coefficient |

**Table A2: JULES sea ice namelist updated from Ridley et al. (2018)**