# Peer review of "The sea ice component of GC5: coupling SI3 to HadGEM3 using conductive fluxes"

_EGUsphere, 2023_

## Author Comment (AC1)

**REVIEW #2 RESPONSE**

We thank Reviewer #2 for their constructive review and for raising some interesting points for us to consider. Here we address each of the reviewer's comments in turn; in the ensuing text the *reviewer comments are provided in italics* with author responses provided in blue.

*The authors present a detailed description and limited evaluation of GSI9, an upgrade to their sea ice coupling scheme for UKESM. While their previous GC3.1/CMIP6 setup employed CICE in both the ocean model NEMO and the atmosphere & land model UM-JULES, the new GC5 prototype for CMIP7 use replaces CICE on the ocean side with the NEMO native sea ice model SI3. The conductive coupling approach, which places the atmosphere-sea ice coupling interface between the top and second layer of sea ice rather than between the top layer and the atmosphere, remains unchanged.*

*The technical description of the work is clear and concise, and the idiosyncrasies of the conductive coupling approach are well-documented in the cited literature. However, I would like to see a paragraph on the motivation for using SI3 for NEMO4 rather than continuing to use CICE.*

Technically the main focus of the paper is to provide a detailed description of GSI9, an upgrade to the UK sea ice model configuration. This GSI9 configuration, which is part of the GC5 coupled model configuration, will form the physical basis for the UKESM2 and HadGEM3 contributions to CMIP7. We also provide a thorough documentation of the sea ice coupling within GC5. Although the GSI9 configuration has transitioned from using CICE to SI3, which has required adaptation of the latter to work with the coupling, the fundamentals of the conductivity coupling approach is largely unchanged from the previous model versions (except for the addition of penetrating short-wave radiation described in this paper).

Another technical clarification is that, whilst it is true that the previous configuration used CICE on the ocean side of the coupler and the new configuration uses SI3, it is not true that either of the configurations use CICE on the atmosphere side of the coupler. The UM-JULES code used on the atmosphere side of the coupler includes an adapted version of the CCSM3 4-band albedo scheme (as outlined in Collins et al., 2006) that was previously used in CICE. However, we do not use CICE per se.

**We shall modify the introductory text to ensure that these points are made more clearly.**

Regarding the motivation for using SI3 with NEMO 4 rather than 'continuing to use CICE' CICE we should make the point that this would have meant staying with the CICE5 codebase, which is not being actively developed anymore. Our conductivity coupling changes were not included in the CICE5 code that was used to initiate CICE6 development, and subsequently changed considerably. This means that we would have to have undertaken considerable development to 'move' to CICE6, along the same lines as moving to SI3 (or move away from the implicit coupling formulation that we use in the MetUM).

The choice to use SI3 was more about strategic and technical efficiencies than scientific. The scientific complexity of the SI3-based sea ice model configuration described in this paper (GSI9) is broadly similar than the CICE-based sea ice model configuration (GSI8)

used previously. There are some slight differences (e.g. the lateral melting and salinity evolution are more advanced in GSI9, the sea ice strength was more advanced in GSI8), which are described in this paper already.

Meanwhile SI3 is part of the NEMO ocean modelling framework, which is the UK's ocean modelling framework of choice. As NEMO consortium members we therefore physically own SI3. We are also involved already in the management and technical development of the NEMO code and so there are economies of scale there. From a technical point-of-view, the fact that SI3 runs on the same grid as the NEMO ocean component also provides improvements. The efficiency of the ocean-sea ice coupling is greatly increased rather than having to interpolate velocity fields twice on every model time-step – both before, and after, the sea ice model is called. More importantly there are also scientific differences between the NEMO ocean and CICE sea ice grid can cause problems for the advection of sea ice in narrow channels. In the previous model component this disparity would often lead to the creation of ice pillars in areas of tight bathymetry where sea ice was static but the ocean mobile.

**We will try to work some of these points into the paper introduction, noting that not all of the above discussion is relevant for a configuration description or suitable for inclusion in a scientific paper.**

References:

Collins, W. D., and Coauthors, 2006: The Community Climate System Model Version 3 (CCSM3). J. Climate, 19, 2122–2143, https://doi.org/10.1175/JCLI3761.1.

*Additionally, the manuscript would benefit from detailing why the advantages of using SI3 outweigh any potential advantages of maintaining a consistent sea ice physics formulation with CICE on both sides of the coupler.*

As stated above, we do not use CICE on the atmosphere side of the coupler. The radiation and surface exchanges are carried out within the JULES model and not within CICE. The CCSM3 radiation scheme used in JULES is based upon that used within CICE, but that is independent of the choice of sea ice model used and no other aspects of the surface exchange are related to CICE. Therefore, there are no "potential advantages of maintaining a consistent sea ice physics formulation with CICE on both sides of the coupler".

In fact, the opposite is true because with SI3 the sea ice is now on the same grid as the ocean, which has removed one of the inconsistencies that we had in the ocean-sea ice coupling. Furthermore, the conductivity coupling itself already removes the largest inconsistency that exists in standard coupling approach by allowing a consistent calculation of surface exchanges and (atmospheric) boundary-layer evolution across the globe. When using the standard coupling approach, the sea ice-atmosphere interface is actually located above the sea ice and bulk formulae are used to calculate the surface exchanges either side of the coupler (often not using the same bulk formulae).

**We will tighten up the text in the introduction and the conductivity coupling description in Section 3 to make the above points clearer.**

*Although the authors state that a detailed analysis of the resulting sea ice climate is not within the scope of this paper, I encourage them to broaden the scope slightly to include a basic characterization of the sensitivity of the old and new sea ice schemes different climate states. One approach could be to run a short 1850 control simulation followed by a 1% CO2 increase per year, allowing for the computation of transient climate response (TCR) and sea ice response. Another option may be to approximate the CMIP6 HighResMIP protocol (as forcing implementation permits) and show the Arctic Amplification Indices and sea ice response. Other approaches can yield similar information. I consider this relevant as UKESM GC3.1 was an outlier with the highest climate sensitivity CMIP6 dataset.*

The reviewer is correct to highlight that climate sensitivity is a pertinent issue for the CMIP climate model contributions. Whilst the GC5 configuration will form the physical basis for the UK's CMIP7 contributions, it will not be used 'out of the box'. Development of the UK CMIP7 climate model configurations from GC5 (inc. UKESM2) is an active area of research – both at the Met Office and within the wider UKESM group. There will likely be several papers on this activity produced in the build-up to our releasing CMIP7 simulations, including an overview of the climate model performance. It would therefore be misleading for us to include details of climate sensitivity or transient climate evolution in this paper.

*Finally, it would be beneficial to include a discussion, or provide a reference if analyzed elsewhere, on the phase error in the Arctic summer sea ice area minimum, which occurs in September in observations but shifts to August in both UKESM versions., See Figure 2. The improvements in the Southern Hemisphere, on the other hand, are very encouraging, even if much of it may be the result of ocean modelling improvements.*

Regarding the August minimum in Arctic sea ice area: it is worth noting that there are considerable uncertainties associated with the satellite observations in the high summer, particularly August, given that the presence of surface melt water (melt-ponds) increases the relative error of passive microwave observations considerably. The passive microwave retrieval algorithms artificially inflate the sea ice concentrations to compensate for the presence of melt ponds, leading to considerably higher uncertainties in the observational products during the summer months. For these reasons many previous studies have used extent instead of area. However, this is not considered good practice for sea ice studies because extent is a nonlinear, grid-dependent metric – see the arguments presented in SIMIP Community (2020) & Notz (2014).

This difference in phase between the satellite observations and model is not a new thing and is not unique to UK model versions. If we used extent instead of area, as reported for previous configuration, then the modelled seasonal minimum would be in September and consistent with satellite observations. It should also be noted that an August areal minimum is certainly not unique to the UK models. Figure 1b of Roach et al. (2020) (replicated as Figure 1 below) shows that around a quarter of the CMIP6 models considered in that study (at least 10 of 40) have lower sea ice area in August then September. Meanwhile Keen et al. (2021) found the same to be true for around half of the subset of models they considered (7 of 15).

[Figure]

*Figure 1: Arctic sea ice area mean seasonal cycles from 40 CMIP6 models (thin green lines) along with multi-model mean (thick green dashed line) and three observational references (thick black lines). Taken from Roach et al. (2020) Figure 1b.*

All that said, the reviewer is correct that this is an interesting topic. The fact that the minimum extent occurs in September but minimum area in August suggests a competition between melting of the ice edge at lower latitudes and (re)freezing processes in the ice pack at higher latitudes. After all, changes in area within the central ice pack would not contribute to increased extent if the concentration were already above the 15% extent threshold.

This competition between growth and melt is illustrated in Figure 2 below, which shows the average total sea ice mass flux due to thermodynamic processes for August, September & October. Whilst August is entirely melting (green) and October is dominated by growth (purple), the situation in September is a balance between ice melt at lower latitudes and ice growth at higher latitudes. For the model it seems that the growth processes out-weight the melting to produce a higher area in September than August.

**We shall add a paragraph to Section 4 to explain the above.**

It would be really interesting to dig into this further and ascertain at what point in September the model changes from net melt to net growth. It might be that the model timing is only out by a few days and that this is biasing the September mean. However, we unfortunately only have monthly output because storing daily fields from these 100-year coupled model runs would be a challenge. This will have to remain something interesting to investigate in future.

[Figure]

*Figure 2: illustration of the competing thermodynamic contributions to Arctic sea ice melt/growth in September. Showing the total sea ice mass change due to thermodynamic processes in Augst, September, and October averaged over the 50-year assessment period used in this paper. Positive fluxes (purple) represent sea ice growth, whilst negative (green) fluxes represent sea ice melt.*

References:

Keen, A., Blockley, E., Bailey, D. A., Boldingh Debernard, J., Bushuk, M., Delhaye, S., Docquier, D., Feltham, D., Massonnet, F., O'Farrell, S., Ponsoni, L., Rodriguez, J. M., Schroeder, D., Swart, N., Toyoda, T., Tsujino, H., Vancoppenolle, M., and Wyser, K.: An inter-comparison of the mass budget of the Arctic sea ice in CMIP6 models, The Cryosphere, 15, 951–982, https://doi.org/10.5194/tc-15-951-2021, 2021.

Notz, D.: Sea-ice extent and its trend provide limited metrics of model performance, The Cryosphere, 8, 229–243, https://doi.org/10.5194/tc-8-229-2014, 2014.

Roach, L. A., Dörr, J., Holmes, C. R., Massonnet, F., Blockley, E. W., Notz, D., et al.: Antarctic sea ice area in CMIP6. Geophysical Research Letters, 47, e2019GL086729, https://doi.org/10.1029/2019GL086729, 2020.

SIMIP Community: Arctic sea ice in CMIP6, Geophysical Research Letters, 47, e2019GL086749. https://doi.org/10.1029/2019GL086749, 2020.

*Overall, I recommend the paper receive a minor revision before acceptance.*

Great! Many thanks for the constructive review and for raising some interesting points.

*Minor Comments:*

- *L525: Why is second-order accuracy used for only two out of the nine radiative fluxes?*
  We prefer to use second order regridding where possible because it is higher accuracy. However, this is not possible for spatial fields that contain high horizontal gradients or heterogeneity because the second order schemes can cause over-shoots. This could lead to unphysical quantities in the coupling fields such as negative radiation or an ice area fraction outside of the range [0,1].

- *Appendix A, page 24: Snow volume is halved after ridging (as noted in the text), yet the melt pond fraction remains the same (rn_fpndrdg=1). While this may be good for tuning, it seems incorrect from a practical perspective. If the snow falls off, so should the liquid water.*
  This is an interesting point although not one we can address in this paper or as part of this model configuration, which is already frozen. We are currently performing some runs to test the sensitivity of the model configuration to the proportion of melt-ponds retained during ridging to inform the parameter settings used in our next model configuration.

- *Appendix A, page 24: The text mentions increasing the number of layers from 2 to 4, but here it says the layers are reduced from 4 to 2. Please verify this information.*
  On page 24 the namelist table states that we use 4 layers, but that the default for the SI3 model is 2. Therefore, we have increased the number of layers from the default of 2 up to 4, which is consistent with the text, which mentions increasing the number of layers from 2 to 4.

---

## Author Comment (AC2)

**REVIEW #1 RESPONSE**

We thank Reviewer #1 for their constructive review and for raising some interesting points for us to consider. We apologise to the reviewer for the considerable amount of time that has elapsed since the original review was submitted last year. This was an artefact of the fact that the discussion period was delayed several times and so the 2nd review was only received in May this year.

Here we address each of the reviewer's comments in turn; in the ensuing text the *reviewer comments are provided in italics* with author responses provided in blue.

*In this paper, Ed Blockley and his co-authors provide a description of the sea-ice model GSI9 and its coupling within the Coupled Model GC5. The paper is clearly structured, provides a helpful summary of the functioning of the model, describes in detail the technical aspect of the coupling of GSI9 to the ocean and to the atmosphere, and is very well written. I therefore generally recommend its publication, but I think that the following comments should be addressed in a revised version. They are in particular geared towards increasing the usefulness of this paper for a general sea-ice modelling audience who would like to draw insights from such paper for their own work.*

Great, many thanks for the constructive review. We feel we should point out here that the focus of the paper is not to provide a description of a sea ice model, but rather a sea ice model **configuration** (GSI9). The distinction is an important one because a model configuration, although based upon the model itself, is more about how the model is used than the model itself. In this manuscript we provide a technical documentation of the GSI9 sea ice model configuration that will form the sea ice component for the UK's model contributions to CMIP7 (HadGEM3-GC5 & UKESM2). Although we hope the paper would be of interest to the sea ice modelling community, that is not the primary aim of the paper. The aim is to have a documentation of the sea ice component that can be cited by anyone using HadGEM3-GC5/UKESM2 or the CMIP7 data produced by them.

**We shall review the introductory text to ensure that this message is clearly made.**

*Overall, I felt that the paper provides too little guidance to the reader regarding the motivation of the shift to SI3 and to the new coupling scheme. A description of the advantages and disavantages of SI3 vs CICE (if possible) and of flux coupling vs. standard coupling would be helpful. The reader could then infer for themselves whether such shift is considered useful for scientific / numerical / strategic reasons.*

Re. the 'new coupling scheme': Although the GSI9 configuration has transitioned from using CICE to SI3, which has required adaptation of the latter to work with the coupling, the fundamental formulation of the conductivity coupling scheme used here is largely unchanged from the previous model versions (except for the addition of penetrating short-wave radiation described in this manuscript). Use of the conductivity coupling approach allows a consistent calculation of surface exchanges and (atmospheric) boundary-layer evolution across the globe. When using a standard coupling approach, the sea ice-atmosphere interface is actually located above the sea ice and bulk formulae are used to calculate the surface

exchanges either side of the coupler. Often this is done using different bulk formulae for the ocean/sea ice as in the atmosphere.

The advantages associated with the conductivity coupling have been covered in previous papers including the West et al. (2016) and Ridley et al. (2018) papers cited in this manuscript. Meanwhile the framework for implicit coupling methods such as this are outlined in Best et al. (2004).

**We shall improve the introduction and discussion of the conductivity coupling to better explain the findings of West et al. (2016) and include reference to Best et al. (2004).**

Re. the 'shift to SI3': we shall start by making the point that continuing to use CICE would have meant staying with the CICE5 codebase, which is not being actively developed anymore. Our conductivity coupling changes were not included in the CICE5 code that was used to initiate CICE6 development, and subsequently changed considerably. This means that we would have to have undertaken considerable development to 'move' to CICE6, along the same lines as moving to SI3 (or move away from the implicit coupling formulation that we use in the MetUM).

The choice to use SI3 was more about strategic and technical efficiencies than scientific. The scientific complexity of the SI3-based sea ice model configuration described in this paper (GSI9) is broadly similar than the CICE-based sea ice model configuration (GSI8) used previously. There are some slight differences (e.g. the lateral melting and salinity evolution are more advanced in GSI9, the sea ice strength was more advanced in GSI8), which are described in this paper already.

Meanwhile SI3 is part of the NEMO ocean modelling framework, which is the UK's ocean modelling framework of choice. As NEMO consortium members we therefore physically own SI3. We are also involved already in the management and technical development of the NEMO code and so there are economies of scale there. From a technical point-of-view, the fact that SI3 runs on the same grid as the NEMO ocean component also provides improvements. The efficiency of the ocean-sea ice coupling is greatly increased rather than having to interpolate velocity fields twice on every model time-step – both before, and after, the sea ice model is called. More importantly there are also scientific differences between the NEMO ocean and CICE sea ice grid can cause problems for the advection of sea ice in narrow channels. In the previous model component this disparity would often lead to the creation of ice pillars in areas of tight bathymetry where sea ice was static but the ocean mobile.

**We will try to work some of these points into the paper introduction, noting that not all of the above discussion is relevant for a configuration description or suitable for inclusion in a scientific paper.**

However, we should point out that the advantages and disadvantages of CICE vs SI3 (whether that be staying with CICE5 or shifting to CICE6) will be largely unique to our model setup and the specifics of the configurations that we run. There are so many different physical choices available in either CICE or NEMO/SI3 that it can be very misleading to talk about the advantages of one versus the other out of context.

Whilst we acknowledge that an in-depth comparison of the CICE and SI3 models (and indeed any other models) would be useful for the sea ice modelling community, we note that it would indeed be a sizeable undertaking. Such a comparison would require evaluation would require a series of tightly defined numerical experiments to understand the model

performance in various situations and a thorough exploration of how the models perform across the complexity and parameter space. This is beyond the scope of this study, which is primarily a documentation of our CMIP7 sea ice model component to provide a technical reference for others using the model or data from it.

*I also would have liked to see a broader discussion of the performance of the new model setup. How computationally expensive are these runs relative to the ones with the previous model version (or: Which percentage of the ocean computations happen within the sea-ice module in this version and in the previous version)? How well does the sea-ice model scale in this setup with the number of CPUs compared to the previous model version, unless this is documented elsewhere?*

The, somewhat dull, answer to this question is that the computational performance of the new GSI9 configuration (based upon SI3) is almost the same as the old GSI8 configuration (based upon CICE). This is not a great surprise given that the two configurations are broadly comparable in their scientific complexity and the underlying models are based upon the same structural formulation - being dynamic-thermodynamic viscous-plastic continuum sea ice models along the lines of the pioneering work of Hibler (1979) and the AIDJEX group (Coon et al., 1974) (see discussions in Blockley et al, 2020). A thorough exploration of the scalability and relative costs of the new and old configuration is beyond the scope of this study

For the coupled model configurations used in this study – both the new GC5 and the old GC3 configuration – the sea ice uses such a small fraction of the total computational cost or run-time, that we have not looked in any great depth at the relative costs of the two sea ice components within the coupled model. Whilst performing the initial testing of the SI3 configuration we found the run-time to be similar to the CICE model. Any differences between in run-time between the two sea ice versions is small enough to be lost in the noise related to other changes to the system.

To provide an illustration, we have analysed the run-time of the ocean-sea ice tasks in the forced version of the GSI9 & GSI8 model configurations (as introduced in Guiavarc'h et al., 2024). The old system using GSI8 (CICE) performs one calendar month in approximately 65 minutes on 16 nodes of our HPC. Meanwhile the new system using GSI9 (SI3) performs one calendar month in approximately 41 minutes on 12 nodes of our HPC. On the face of it the new system seems a lot quicker and cheaper. Some of this speed-up comes from the fact that the ocean-sea ice timestep has been increased from 22.5 minutes to 30 minutes. However, even accounting for the fact that a factor of 1.33 less work is being done by the new configuration, it is still faster (41 * 1.33 = ~55 mins) and this is using fewer compute nodes. Some of this efficiency improvement will have come from the fact we no longer have to couple (interpolate) between NEMO and CICE grids at each time-step.

References:

Blockley, E., and Coauthors, 2020: The Future of Sea Ice Modeling: Where Do We Go from Here?. Bull. Amer. Meteor. Soc., 101, E1304–E1311, https://doi.org/10.1175/BAMS-D-20-0073.1.

Coon, M. D., G. A. Maykut, R. S. Pritchard, D. A. Rothrock, and A. S. Thorndike, 1974: Modeling the pack ice as an elastic-plastic material. AIDJEX Bull., No. 24, University of Washington, Seattle, WA, 105 pp

Hibler, W. D., 1979: A dynamic thermodynamic sea ice model. J. Phys. Oceanogr., 9, 815–846, https://doi.org/10.1175/1520-0485(1979)009<0815:ADTSIM>2.0.CO;2

*For the coupling, more information would be helpful regarding the standard coupling frequency and standard time step for the atmosphere and the ocean. I was surprised to read that coupling occurs at 2-3 ocean and atmosphere timesteps, as this seems to imply a similar time step in the atmosphere and the ocean. Is this indeed the case? Most models that I am aware of use a much longer time step in the ocean than in the atmosphere.*

The implication that ocean and atmosphere time-steps are similar is a little surprising. The relative similarity of the time-step length comes, in part, from the fact that the ocean is run at relatively higher resolution than the atmosphere for reasons of computational cost. For the N216-ORCA025 configuration used to generate the model outputs in this paper the nominal resolution of the atmosphere is 100 km with time-step of 15 minutes. Meanwhile the nominal resolution of the ocean is 25 km and the time-step for GC5 (GC3.1) is 30 (20) minutes. So for GC5 configuration at N216-ORCA025 resolution the components couple every hour, which for the ocean is every $2^{nd}$ timestep and for the atmosphere is every $4^{th}$ timestep.

Table 2 of Roberts et al. (2019) provides details of the spatial resolution and time-step used for all the different model resolutions of HadGEM3 that we contributed to CMIP6. It is worth noting that these values are given for GC3.1 but are almost all correct for GC5. The only change is that the ocean time-step is now longer (30 mins for ORCA025 instead of 20 mins) owing to increased stability of the model using the semi-implicit sea ice drag (see Guiavarc'h et al., in review).

**We will tidy up this wording in Section 3. The "2-3" shall be changed to "between 2 and 4" and we shall elaborate a little further to avoid confusion.**

References:

Guiavarc'h, C., Storkey, D., Blaker, A. T., Blockley, E., Megann, A., Hewitt, H. T., Bell, M. J., Calvert, D., Copsey, D., Sinha, B., Moreton, S., Mathiot, P., and An, B.: GOSI9: UK Global Ocean and Sea Ice configurations, EGUsphere [preprint], https://doi.org/10.5194/egusphere-2024-805, 2024.

*Are there any drawbacks for a lower coupling frequency in the flux-coupling approach compared to the standard approach, given the somewhat unphysical development of internal sea-ice temperature for a fixed surface temperature between the coupling intervals? Or is the flux coupling of advantage, as the surface temperature and the atmosphere interact physically more realistically, and this is considered more important in a coupled setup?*

The whole ethos of the conductivity coupling approach is to include the rapidly changing ice surface temperature into the atmosphere where the timescales are shorter. This allows the near-surface atmosphere to respond instantly to changes in the ice surface temperature and vice versa. The more slowly evolving internal ice temperature then responds on the slightly longer coupling timescale (West et al., 2016; Ridley et al., 2018). Although there will still be a lag in the system associated with the coupling period, in the conductivity coupling that lag will exist lower down in the sea ice where the processes are already slower moving.

As the reviewer speculates this leads to a more physically realistic atmosphere-ice surface interaction in the coupled model. It also allows us to undertake a consistent calculation of surface exchanges and near-surface boundary-layer across the whole atmosphere model (see Best et al., 2004).

There are no drawbacks to a lower coupling frequency in the conductivity coupling approach. In fact the opposite is true because the benefits of the conductivity coupling approach are more marked with a lower coupling frequency (longer coupling period). You can see this in the analysis of West et al. (2016) at https://doi.org/10.5194/gmd-9-1125-2016. They show the errors in surface heat flux and surface temperature seen by the atmosphere for 3-hourly coupling frequency in Figure 5b,c (respectively) are much higher than for 1-hourly coupling (equivalent panels in Figure 4b,c).

**We shall improve the introduction and discussion of the conductivity coupling to better explain the findings of West et al. (2016) and include reference to Best et al. (2004).**

References:

Best, M. J., Beljaars, A., Polcher, J., and Viterbo, P.: A Proposed Structure for Coupling Tiled Surfaces with the Planetary Boundary Layer, J. Hydrometerorol., 5, 1271–1278, doi:10.1175/JHM382.1, 2004.

*Regarding model evaluation, the current analysis does not allow too much insight regarding the source of the potential model improvement, and in particular little insight into potential improvements related to the sea-ice component. For the Southern Ocean, improvements in the ocean component rather than in the sea-ice component are mentioned as the primary reason for the improved sea-ice simulation. It remains unclear whether improvements e.g. in the atmosphere component are the primary reason for the improved sea-ice thickness distribution in the Arctic. Is there any indication that the sea-ice component itself has contributed to these improvements in either hemisphere? I know this is hard to show, but maybe an analysis of more sea-ice inherent properties would be more helpful (e.g., melt-pond fraction, small-scale variability, lead fraction, or the like).*

The scope of this paper is to document the sea ice configuration and coupling in the latest UK model configuration, which will form the basis of the upcoming contributions to CMIP. As such we are concerned with documenting the sea ice coupling and the scientific options that comprise the GSI9 configuration, and to demonstrate that the GC5 sea ice simulations are realistic. We include comparison to the previous version used for CMIP6 as part of the demonstration of realistic results. However, the primary focus of this work is not to provide a thorough attribution of these changes – just to show a top-level overview.

Although we want to keep the focus on the configuration itself, there may be scope to include a little more analysis without diluting the main message of the paper. **We shall explore this and make changes as appropriate**, with a focus on the Arctic ice thickness distribution shown in Figure 3. However, as the reviewer notes, it is very difficult to unpick cause and effect when dealing coupled model feedbacks in such a complex system.

*Finally, I was not sure why HadISST was used for the sea-ice concentration comparison rather than one of the standard direct satellite products. Using HadiSST 2.0 can be*

*misleading, as they use a very generous ocean grid with many islands etc being water. "This results in much larger sea ice extents in HadISST.2 for all calendar months, unless the same mask is applied. We recommend that the same grid and mask are used when comparing any sea ice concentration data set." (quote from HadISST 2.2.1.0 website)*

The HadISST.2 dataset is used because it provides a consistent time-series of sea ice concentration against which to compare the model output. HadISST.2 is derived from the same satellite data used for the standard products but it has been biased corrected using ice charts, which provide a better estimate of sea ice (certainly at low concentrations and during the summer months).

The key reason for using HadISST.2 however is because it allows us to correctly "compare apples with apples". As the reviewer notes, the HadISST.2 data is provided on a generous ocean grid. Doing so allows the user to specify their own land-sea mask rather than having to accept whatever was used by whichever group performed the processing and/or extent index calculation (e.g. NSIDC, OSI SAF). The motivation for this is precisely so that any inconsistencies related to the land-sea mask can be overcome. We therefore know that the area calculations for both the model and observations are being performed with the same land-sea mask.

*Minor comments:*

- *l.69: What does "largely similar" mean? Shouldn't it be either "similar" or "largely equal"? ;)*
  In English "largely similar" is a phrase used to compare two similar things or ideas, suggesting that there are some differences but the overall nature is the same (see https://ludwig.guru/s/largely+similar+to). It's a more nuanced way to say "similar" but we can drop the "largely" if that is likely to cause confusion.

- *l.76: Are the uppermost ocean grid cells completely turned into ice as the ice approaches very large thickness? Which coordinate system is used in the ocean model and how is the ice incorporated into it?*
  The default option for sea ice in NEMO, as for many ocean models, is that the sea ice levitates above the ocean and so has no impact on the ocean vertical coordinate. The current version of the ocean (GOSI9; Guiavarc'h et al., in review). The model uses a z-level vertical coordinate with 75 vertical levels and partial-step topography. The level thickness is double tanh function of depth increasing from 1m thickness near the surface to 200m at 6000m depth. **We shall include a sentence to explain that the sea ice levitates above the ocean rather than being embedded in it.**

- *l.89: Is this a different convection scheme compared to previous model versions? If so, what was used before? Is third-order necessary and/or helpful?*
  We presume you mean "advection" rather than "convection" here?  If so the advection scheme is the default in the SI3 model but different from that used in the CICE version. Re. "third order": the Prather advection scheme is actually second-order not third-order and this was a typo. **This typo will be corrected.**

- *l.96: Not sure what "this" refers to*
  This is the formulation of Flato and Hibler (1995) mentioned in the previous sentence.

**We shall change the wording to make that clearer.**

- *l.128: Would be helpful to briefly indicate how the heat-flux calculation by Maykut and McPhee (1995) works*
  *OK.* **We shall add a sentence here.**

- *l.136: What is "the beta coefficient"?*
  The beta coefficient is used in the lateral melting scheme as part of the floe diameter parameterisation. It is akin to the \beta used in equations 26-27 of Lüpkes et al. (2012). **We will add a sentence to better explain this.**

- *l.147: Is salinity also used to calculate energy content / heat capacity in the thermodynamics?*
  The salinity is used to calculate all thermal properties, which includes specific heat, thermal conductivity, enthalpy, and freezing/melting temperatures – see Vancoppenolle et al. (2009), Rousset et al. (2015) or the SI3 documentation (Vancoppenolle et al., 2023).
  **The text at the end of Section 2.3 that says salinity is used to calculate sea ice conductivities in the thermodynamics shall be modified to include the above information.**

- *l.226: How relevant is passing these velocities to the atmosphere, as they should be relatively small compared to the wind speed. Is the numerical overhead of passing them over negligible? Or would setting them to 0 in the atmosphere suffice? (Maybe nothing to be examined for this paper, but maybe this is known)*
  In our coupled model configuration the default behaviour is to calculate momentum and scalar exchanges using relative velocities. In this regard the sea ice velocities are used with the ocean currents to calculate ocean-ice exchange, the ocean velocities are used with the winds to calculate ocean-atmosphere exchange, and the sea ice velocities are used with the winds to calculate atmosphere-ice exchanges. Not doing this would feel very wrong and so we have not tried setting them to zero in the atmosphere. Within the coupling the ocean and sea ice velocities are combined and passed through the coupler together (see Table 1) and so the numerical overhead of including the sea ice velocities in the surface exchange is almost negligible.

- *l.302ff: I would recommend to leave this out, or to move it to somewhere else. It's neither a conclusion, nor really helpful at this stage, I find. Maybe it'd be better to integrate the individual future plans into the relevant sections of the paper, with a brief motivation.*
  It is quite normal to include future plans at the end of system description paper such as this, but they can easily be removed if the reviewers would prefer this. However we wouldn't want to integrate them into the relevant sections of the because we are concerned about misleading conclusions being drawn by the reader. It is better to maintain a clear split between what is in the current configuration and what are the future plans.

References:

Lüpkes, C., V. M. Gryanik, J. Hartmann, and E. L. Andreas (2012), A parametrization, based on sea ice morphology, of the neutral atmospheric drag coefficients for weather prediction and climate models, J. Geophys. Res., 117, D13112, https://doi.org/10.1029/2012JD017630.

*Typos / Grammar:*

- *l.47: Drop "is"*
  Agreed. We shall make this change.

- *l.48: "are discussed in the GC5 paper"*
  Agreed. We shall make this change.

- *l.80: Drop comma*
  There is no comma on line 80 but there is one on line 79 that we shall drop.

- *l.90: 'or "to" open water' is somewhat easier to read, I find*
  We're not sure this would really help. This sentence is a bit of a mouthful and so we shall reword it.